# Kernel methods through the roof: handling billions of points efficiently

**Giacomo Meanti**
MaLGa, DIBRIS
Università degli Studi di Genova
giacomo.meanti@edu.unige.it

**Luigi Carratino**
MaLGa, DIBRIS
Università degli Studi di Genova
luigi.carratino@dibris.unige.it

**Lorenzo Rosasco**
MaLGa, DIBRIS, IIT & MIT
Università degli Studi di Genova
lrosasco@mit.edu

**Alessandro Rudi**
INRIA - École Normale Supérieure
PSL Research University
alessandro.rudi@inria.fr

## Abstract

Kernel methods provide an elegant and principled approach to nonparametric learning, but so far could hardly be used in large scale problems, since naïve implementations scale poorly with data size. Recent advances have shown the benefits of a number of algorithmic ideas, for example combining optimization, numerical linear algebra and random projections. Here, we push these efforts further to develop and test a solver that takes full advantage of GPU hardware. Towards this end, we designed a preconditioned gradient solver for kernel methods exploiting both GPU acceleration and parallelization with multiple GPUs, implementing out-of-core variants of common linear algebra operations to guarantee optimal hardware utilization. Further, we optimize the numerical precision of different operations and maximize efficiency of matrix-vector multiplications. As a result we can experimentally show dramatic speedups on datasets with billions of points, while still guaranteeing state of the art performance. Additionally, we make our software available as an easy to use library[1].

## 1   Introduction

Kernel methods provide non-linear/non-parametric extensions of many classical linear models in machine learning and statistics [45, 49]. The data are embedded via a non-linear map into a high dimensional feature space, so that linear models in such a space effectively define non-linear models in the original space. This approach is appealing, since it naturally extends to models with infinitely many features, as long as the inner product in the feature space can be computed. In this case, the inner product is replaced by a positive definite kernel, and infinite dimensional models are reduced to finite dimensional problems. The mathematics of kernel methods has its foundation in the rich theory of reproducing kernel Hilbert spaces [47], and the connection to linear models provides a gateway to deriving sharp statistical results [53, 10, 54, 6, 4, 56]. Further, kernel methods are tightly connected to Gaussian processes [40], and have recently being used to understand the properties of deep learning models [23, 29]. It is not a surprise that kernel methods are among the most theoretically studied models. From a numerical point of view, they reduce to convex optimization problems that can be solved with strong guarantees. The corresponding algorithms provide excellent results on a variety of data-sets, but most implementations are limited to problems of small/medium size, see

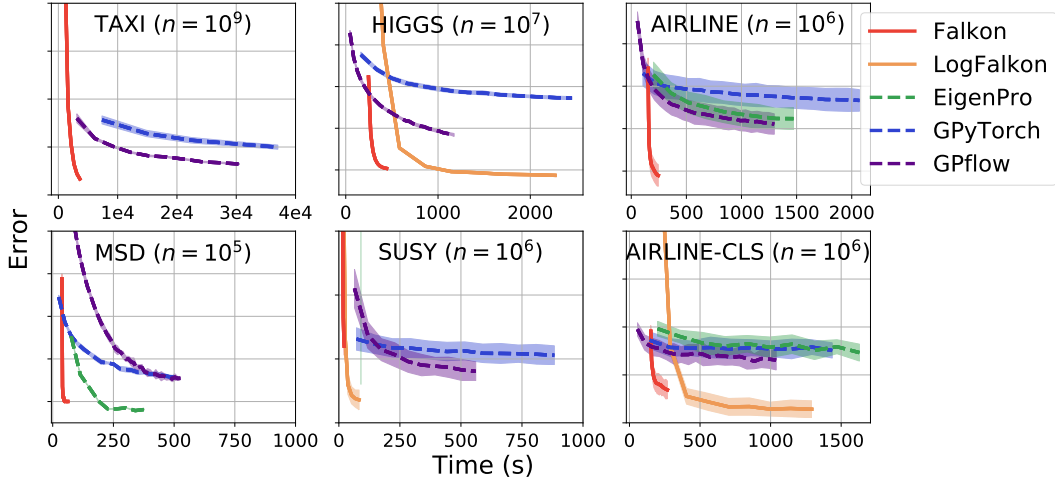

Figure 1: Benchmarks of kernel solvers on large scale datasets with millions and billions points (see Section 4). Our approach (red and yellow lines) consistently achieves state of the art accuracy in minutes.

discussion in [52], Chapter 11. Most methods require handling a kernel matrix quadratic in the sample size. Hence, dealing with datasets of size $10^4$ to $10^5$ is challenging, while larger datasets are typically out of reach. A number of approaches have been considered to alleviate these computational bottlenecks. Among others, random features [38, 39, 66, 26, 12, 11] and the Nyström method are often used [61, 50], see also [14, 25, 18, 3, 67, 9]. While different, both these approaches consider random projections to reduce the problem size and hence computational costs. Renewed interest in approximate kernel methods was also spurred by recent theoretical results proving that computational gains can possibly be achieved with no loss of accuracy, see e.g. [27, 55, 41, 4, 42, 5].

In this paper, we investigate the practical consequences of this line of work, developing and testing large scale kernel methods that can run efficiently on billions of points. Following [43] we use a Nyström approach to reduce the problem size and also to derive a preconditioned gradient solver for kernel methods. Indeed, we focus on smooth loss functions where such approaches are natural. Making these algorithmic ideas practical and capable of exploiting the GPU, requires developing a number of computational solutions, borrowing ideas not only from optimization and numerical analysis but also from scientific and high performance computing [28, 2, 7]. Indeed, we design preconditioned conjugate gradient solvers that take full advantage of both GPU acceleration and parallelization with multiple GPUs, implementing out-of-core variants of common linear algebra operations to guarantee optimal hardware utilization. We further optimize the numerical precision of different operations and investigate ways to perform matrix-vector multiplications most efficiently. The corresponding implementation is then tested extensively on a number of datasets ranging from millions to billions of points. For comparison, we focused on other available large scale kernel implementations that do not require data splitting, or multiple machines. In particular, we consider Eigenpro [30] which is an approach similar to the one we propose, and GPyTorch [16] and GPflow [58] which come from the Gaussian process literature. While these latter solutions allow also for uncertainty quantification, we limit the comparison to prediction. We perform a systematic empirical evaluation running an extensive series of tests. Empirical results show that indeed our approach can process huge datasets in minutes and obtain state of the art performances, comparing favorably to other solutions, both in terms of efficiency and accuracy. More broadly, these results confirm and extend the observations made in [29, 30], that kernel methods can now be seamlessly and effectively deployed on large scale problems. To make these new solutions readily available, the corresponding code is distributed as an easy to use library developed on top of PyTorch [36].

The rest of the paper is organized as follows. In Section 2, we provide some background on the considered approaches. In Section 3, we detail the main algorithmic solutions in our implementation, whereas the last section is devoted to assessing the practical advantages.

## 2 Background

Supervised learning is the problem of inferring an input-output function, given finitely many input-output pairs. In statistical learning theory the data $(x_i, y_i)_{i=1}^n$ are assumed to be sampled independently from a probability distribution $\rho$, and a loss function $\ell(y, f(x))$ is fixed measuring the cost of predicting $f(x)$ in place of $y$. The examples we consider are the squared $(y - f(x))^2$ and the logistic loss $\log(1 + e^{-yf(x)})$. Then, a good function $f$ should minimize the expected loss

$$L(f) = \int \ell\big(f(x), y\big) d\rho(x, y). \tag{1}$$

A basic approach to solve the problem is empirical risk minimization, based on the idea of replacing the above expectation with an empirical average. Further, the search of a solution needs to be restricted to a suitable space of hypothesis, a simple example being linear functions $f(x) = w^\top x$. Kernel methods extend this idea by considering a non linear feature map $x \mapsto \Phi(x) \in \mathcal{F}$ and functions of the form $f(x) = w^\top \Phi(x)$. Here $\Phi(x) \in \mathcal{F}$ can be seen as a feature representation in some space of features. The function space $\mathcal{H}$ thus defined is called reproducing kernel Hilbert space [46]. If we denote by $\|f\|_\mathcal{H}$ its norm then regularized empirical risk minimization is given by

$$\hat{f}_\lambda = \underset{f \in \mathcal{H}}{\arg\min} \frac{1}{n} \sum_{i=1}^n \ell\big(f(x_i), y_i\big) + \lambda \|f\|_\mathcal{H}^2, \tag{2}$$

where the penalty term $\|f\|_\mathcal{H}$ is meant to prevent possible instabilities and $\lambda \geq 0$ is a hyperparameter. From a statistical point of view the properties of the estimator $\hat{f}_\lambda$ are well studied, see e.g. [53, 6, 48]. Under basic assumptions, for $\lambda = \mathcal{O}(1/\sqrt{n})$, it holds with high probability that

$$L(\hat{f}_\lambda) - \inf_{f \in \mathcal{H}} L(f) = \mathcal{O}\left(n^{-1/2}\right). \tag{3}$$

This bound is sharp, but can be improved under further assumptions [6, 53]. Here, we use it for reference. From a computational point of view, the key fact is that it is possible to compute a solution also if $\Phi(x)$ is an infinite feature vector, as long as the kernel $k(x, x') = \Phi(x)^\top \Phi(x')$ can be computed [45]. The Gaussian kernel $\exp(-\|x - x'\|^2/2\sigma^2)$ is a basic example. Indeed, by the representer theorem [24, 46], $\hat{f}_\lambda(x) = \sum_{i=1}^n \alpha_i k(x, x_i)$, so Problem (2) can be replaced with a finite dimensional problem on the coefficients. Its solution depends on the considered loss, but typically involves handling the kernel matrix $K_{nn} \in \mathbb{R}^{n \times n}$ with entries $k(x_i, x_j)$, which becomes prohibitive as soon as $n \sim 10^5$ (although multi-GPU approaches [59] have been recently shown to scale to $10^6$ points). In the following, we focus on Nyström approximation, considering functions of the form

$$f(x) = \sum_{i=1}^m \alpha_i k(x, \tilde{x}_i), \tag{4}$$

where $\{\tilde{x}_1, \ldots, \tilde{x}_m\} \subset \{x_1, \ldots, x_n\}$ are inducing points sampled uniformly at random. As we discuss next, this approach immediately yields computational gains. Moreover, recent theoretical results show that the basic bound in (3) still holds taking as few as $m = \mathcal{O}(\sqrt{n})$ inducing points [42, 31]. With these observations in mind, we next illustrate how these algorithmic ideas can be developed considering first the square loss and than the logistic loss.

**Squared loss.** This choice corresponds to kernel ridge regression (KRR). Since both the loss and penalty are quadratic, solving KRR reduces to solving a linear system. In particular, letting $\boldsymbol{y} = (y_1, \ldots, y_n)$, we obtain $(K_{nn} + \lambda n I)\boldsymbol{\alpha} = \boldsymbol{y}$, for the coefficients $\boldsymbol{\alpha} = (\alpha_1, \ldots, \alpha_n) \in \mathbb{R}^n$ in the solution of the problem in Eq. (2), while using the Nyström approximation (4) we get

$$(K_{nm}^\top K_{nm} + \lambda n K_{mm})\boldsymbol{\alpha} = K_{nm}^\top \boldsymbol{y}, \tag{5}$$

for $\boldsymbol{\alpha} = (\alpha_1, \ldots, \alpha_m) \in \mathbb{R}^m$. The first linear system can be solved directly in $\mathcal{O}(n^3)$ time and $\mathcal{O}(n^2)$ space. In turn, Eq. (5) can be solved directly in $\mathcal{O}(nm^2 + m^3)$ time and $\mathcal{O}(m^2)$ space (if the $K_{nm}$ matrix is computed in blocks). It is well known, that for large linear systems iterative solvers are preferable [44]. Further, the convergence of the latter can be greatly improved by considering preconditioning. The naïve preconditioner $P$ for problem (5) is such that $PP^\top = (K_{nm}^\top K_{nm} + \lambda n K_{mm})^{-1}$, and as costly to compute as the original problem. Following [43] it can be approximated using once again the Nyström method to obtain

$$\tilde{P}\tilde{P}^\top = (\tfrac{n}{m} K_{mm}^2 + \lambda n K_{mm})^{-1} \tag{6}$$

---

**Algorithm 1** Pseudocode for the Falkon algorithm.

1: **function** FALKON($X \in \mathbb{R}^{n \times d}, \boldsymbol{y} \in \mathbb{R}^n, \lambda, m, t$)
2:   $X_m \leftarrow$ RANDOMSUBSAMPLE($X, m$)
3:   $T, A \leftarrow$ PRECONDITIONER($X_m, \lambda$)
4:   **function** LINOP($\boldsymbol{\beta}$)
5:     $\boldsymbol{v} \leftarrow A^{-1}\boldsymbol{\beta}$
6:     $\boldsymbol{c} \leftarrow k(X_m, X)k(X, X_m)T^{-1}\boldsymbol{v}$
7:     **return** $A^{-\top}T^{-\top}\boldsymbol{c} + \lambda n \boldsymbol{v}$
8:   **end function**
9:   $R \leftarrow A^{-\top}T^{-\top}k(X, X_m)\boldsymbol{y}$
10:   $\boldsymbol{\beta} \leftarrow$ CONJUGATEGRADIENT(LINOP, $R, t$)
11:   **return** $T^{-1}A^{-1}\boldsymbol{\beta}$
12: **end function**

13: **function** PRECONDITIONER($X_m \in \mathbb{R}^{m \times d}, \lambda$)
14:   $K_{mm} \leftarrow k(X_m, X_m)$
15:   $T \leftarrow \text{chol}(K_{mm})$
16:   $K_{mm} \leftarrow {}^1\!/_m TT^\top + \lambda \boldsymbol{I}$
17:   $A \leftarrow \text{chol}(K_{mm})$
18:   **return** $T, A$
19: **end function**

Note: LinOp performs the multiplication $\tilde{P}^\top H \tilde{P} \beta$ as in Eq. (8), via matrix-vector products.

---

since $K_{mm}^2 \approx K_{nm}^\top K_{nm}$. Next, we follow again [43] and combine the above preconditioning with conjugate gradient (CG). The pseudocode of the full procedure is given in Algorithm 1. Indeed, as shown in [43] $\mathcal{O}(\log n)$ CG steps are sufficient to achieve the bound in (3). Then with this approach, the total computational cost to achieve optimal statistical bounds is $\mathcal{O}(n\sqrt{n}\log n)$ in time, and in $\mathcal{O}(n)$ in memory, making it ideal for large scale scenarios. The bulk of our paper is devoted to developing solutions to efficiently implement and deploy Algorithm 1.

**Logistic loss.** The above ideas extend to the logistic loss and more generally to self-concordant loss functions, including the softmax loss [32]. For reasons of space, we detail this case in Appendix B and sketch here the main ideas. In this case, iterative solvers are the default option since there is no closed form solution. Nyström method can be used a first time to reduce the size of the problem, and then a second time to derive an approximate Newton step [31]. More precisely, at every step preconditioned conjugate gradient descent is run for a limited number of iterations with a decreasing value of $\lambda$, down to the desired regularization level. In practice, this requires running Algorithm 1 multiple times with small number of iterations $t$ and with decreasing $\lambda$. Making these ideas practical requires efficiently implementing and deploying Algoritm 1, making full use of the available computational architectures. This the core of our contribution that we detail in the next section.

## 3 Reformulating kernel solvers for multi-core/multi-GPU architectures

GPU machines have a peculiar architecture with rather different properties than the standard von Neumann computer, in particular they are characterized by highly parallel computational power, relatively small local accelerator memory and slow memory transfer to/from the accelerator compared to their computational speed [64]. In their standard definition, kernel methods require large amounts of memory with a low density of operations per byte of memory used. This opens the question of how to adapt methods with low operation density to platforms designed to be extremely efficient with very high density of operations per byte. With this in mind, we started considering the state of the art kernel solver with minimal computational requirements for optimal guarantees (described at a high level in Algorithm 1), with the goal to reformulate its computational structure to dramatically increase the density of operations per byte, and reduce as much as possible the required memory use / transfers. To achieve this goal, we use a number of carefully designed computational solutions which systematically reduce the impact of the inherent bottlenecks of multi-core/multi-GPU architectures, while leveraging their intrinsic potential. In particular in the rest of this section we will focus on (a) minimizing the memory footprint of the solver, which has long been the main bottleneck for kernel methods, and is the main limitation encountered by current kernel solvers, (b) dealing with limited

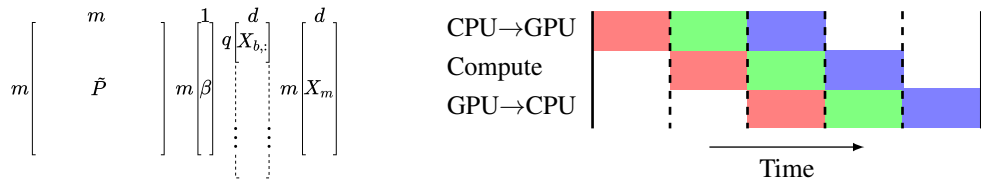

Figure 2: Structure of RAM allocation.   Figure 3: Overlapping memory transfers and computation.

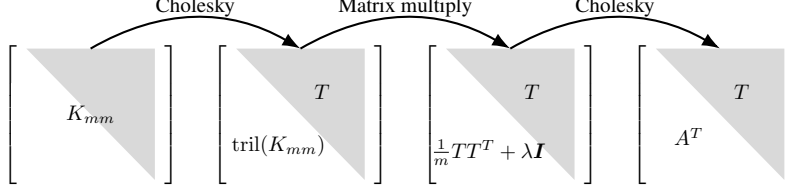

Figure 4: Evolution of the preconditioner matrix in memory.

memory on the GPU, (c) reaching the highest possible accelerator utilization, parallelizing memory transfers and computation, (d) using the enhanced capabilities of GPUs with reduced-precision floating point data.

## 3.1 Overcoming RAM memory bottleneck

Kernel solvers that use the Nyström method need the matrices $K_{mm}$ and $K_{nm}$. Since $K_{nm}$ is used only in matrix-vector products, we can avoid constructing it explicitly (as we shall see in the following paragraphs) which leaves us to deal with the $K_{mm}$ matrix. When $m$ is large, it is crucial to carefully manage the memory needed for this task: in our implementation we only ever allocate one $m \times m$ matrix, and overwrite it in different steps to calculate the preconditioner. Indeed, choosing an appropriate form of the preconditioner, the matrix $K_{mm}$ itself is not needed in the conjugate gradient iteration. Figure 2 shows the total memory usage, which consists of the preconditioner occupying approximately 90% of the memory (see last paragraph of Sect. 3.1), the weight vector $\boldsymbol{\beta}$ and two buffers holding (part of) the $m$ inducing points and a data batch needed to compute $K_{nm}$.

**In-place computation and storage of the preconditioner.** The preconditioner $\tilde{P}$ of Eq. (6) is used to solve a linear system of the form $\tilde{P}^\top H \tilde{P} \boldsymbol{\beta} = \tilde{P}^\top K_{mn} \boldsymbol{y}$ with $H = K_{mn} K_{nm} + \lambda n K_{mm}$ and $\boldsymbol{\beta} = \tilde{P}^{-1} \boldsymbol{\alpha}$. $\tilde{P}$ can be decomposed into two triangular matrices obtained via Cholesky decomposition of $K_{mm}$,

$$\tilde{P} = \tfrac{1}{\sqrt{n}} T^{-1} A^{-1}, \qquad T = \text{chol}(K_{mm}), \qquad A = \text{chol}(\tfrac{1}{m} T T^\top + \lambda \boldsymbol{I}_m). \tag{7}$$

All operations are performed in-place allocating a single $m \times m$ matrix as shown in Figure 4 and as described next: (a) a matrix of dimension $m \times m$ is allocated in memory; (b) the $K_{mm}$ kernel is computed in blocks on the GPU and copied to the matrix; (c) *in-place* Cholesky decomposition of the upper triangle of $K_{mm}$ is performed on the GPU (if the kernel does not fit GPU memory an out-of-core algorithm is used, see later sections); (d) the product $TT^\top$ is computed in blocks via GPU and stored in the lower part; (e) out-of-core in-place Cholesky decomposition is performed on the lower triangle to get $A^\top$. Additional care is needed to take into account the matrix diagonal, not described here for brevity.

**Elimination of the storage of $K_{mm}$.** Considering more carefully the matrix $\tilde{P}(K_{nm}^\top K_{nm} + \lambda n K_{mm})\tilde{P}$ with $\tilde{P}$ as in Eq. (7), we observe that the occurrences of $K_{mm}$ cancel out. Indeed $(T^{-1})^\top K_{mm} T^{-1} = \boldsymbol{I}$ since $K_{mm} = T^\top T$ by Eq. 7. Then, the following characterization allows to overwrite $K_{mm}$ when calculating the preconditioner.

$$\tilde{P}^\top H \tilde{P} \beta = (A^{-1})^\top (T^{-1})^\top (K_{nm}^\top K_{nm} + \lambda n K_{mm}) T^{-1} A^{-1} \boldsymbol{\beta} \tag{8}$$

$$= (A^{-1})^\top [(T^{-1})^\top K_{nm}^\top K_{nm} T^{-1} + \lambda n I] A^{-1} \boldsymbol{\beta}. \tag{9}$$

**Blockwise $K_{nm}$-vector product on GPU.** The conjugate gradient algorithm will repeatedly execute Eq. (9) for different $\boldsymbol{\beta}$. The most expensive operations are the matrix-vector products $K_{nm}^\top(K_{nm}\boldsymbol{v})$ for an arbitrary vector $\boldsymbol{v} \in \mathbb{R}^{m \times 1}$ which – if computed explicitly – would require $n \times m$ memory. However, it is possible to split the input data $X \in \mathbb{R}^{n \times d}$ in $B$ batches of $q$ rows each $\{X_{b,:} \in \mathbb{R}^{q \times d}\}_{b=1}^{B}$, so that matrix-vector products can be accumulated between batches using the formula $\sum_{b=1}^{B} k(X_{b,:}, X_m)^\top (k(X_{b,:}, X_m)\boldsymbol{v})$. The matrix blocks to be held in memory are summarized in Figure 2 for a total size of $m \times (m + d + 1) + q \times d$ where $q$ can be small under memory pressure, or large for greater performance. It is important to note that $k(X_{b,:}, X_m)$ is never stored in main memory, as all operations on it are done on the GPU.

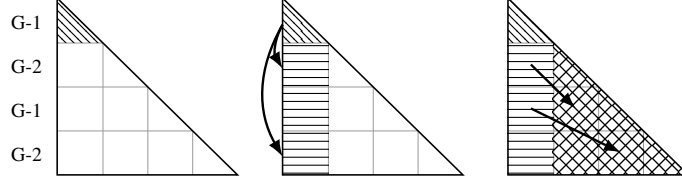

Figure 5: Three phases of the block Cholesky decomposition for updating the first column. Arrows indicate inter-GPU memory transfers between accelerators G-1 and G-2.

## 3.2 Fitting in GPU memory and dealing with multiple GPUs

While the main RAM might be a bottleneck, GPUs have an even smaller amount of memory, and another level of splitting is needed to exploit their speed. For example, a typical architecture has 256GB of RAM and 4 GPUs with 16GB ram each; a preconditioner with $m = 2 \times 10^5$ occupies $150\,\text{GB}$ and $K_{nm}$ with $n = 10^7$ would need $2000\,\text{GB}$ of memory if stored. So we need to deal with both efficient computation of $K_{nm}$-vector product in chunks that fit a GPU, and with the computation of the preconditioner that usually does not fit in GPU memory. Operations based on a large storage layer (main RAM) and a small but fast layer (GPU) are called out-of-core (OOC) operations. However, common machine learning libraries such as Tensorflow [1] or PyTorch [36] do not implement OOC versions of the required matrix operations, leaving potentially complex implementations to the users. Hence, in our library, we provide these implementations in easily reusable form. It is important to note that splitting our workload to fit in GPU also provides an easy path to parallelization in a multi-GPU system: new chunks of computation are assigned to the first free GPU, effectively redistributing the workload between multiple accelerators when available.

**Optimized block decomposition for out-of-core $K_{nm}$-vector multiplication.** As seen in the previous section, matrix-vector products can be split along the dimension $n$, resulting in independent chunks of work that need to be summed up at the end. The OOC product between a kernel matrix and a vector proceeds by: (a) transferring a block of data onto the device, (b) computing the kernel on device and multiplying it by the vector, (c) copying the result back to the host. This sequence of operations minimizes expensive data-transfers between host and device since the kernel matrix is never moved. In particular, the computation is also split along dimensions $m$ and $d$, to maximize the ratio between computational complexity and transfer time: i.e., maximizing $\frac{qrs}{qs+ds}$ subject to $qs + ds \leq G$, where $q$, $r$ and $s$ are the batch dimensions along $n$, $m$ and $d$ respectively, and $G$ is the available GPU memory.

**Out-of-core multi-GPU Cholesky decomposition.** Other operations, such as Cholesky decomposition and triangular matrix multiplication (lines 15, 16, 17 of Algorithm 1), can also benefit from GPU execution. Here we describe, at a high level, our algorithm for multi-GPU OOC Cholesky decomposition inspired by [28, 65]. We leave further details to Appendix C. Consider a symmetric matrix $A$, split into $B \times B$ tiles $A_{ij} \in \mathbb{R}^{t \times t}, i \in [B], j \in [B]$, assumed of equal size for brevity. We want a factorization $A = LL^\top$, where $L$ is lower triangular, with the formula $A_{i,j} = \sum_{k=1}^{j} L_{i,k} L_{j,k}^\top$. The algorithm runs in-place, updating one column of $A$ at a time. Each column update proceeds in three steps, illustrated in Figure 5. Clearly $A_{1,1} = L_{1,1} L_{1,1}^\top$ so we compute $L_{1,1}$ by a Cholesky decomposition on tile $A_{1,1}$ which is small and can be done entirely on the GPU (e.g. with cuSOLVER [34]). Then we consider the other tiles of the first block column of $L$ for which $A_{j,1} = L_{j,1} L_{1,1}^\top$ with $j > 1$. Since we know $L_{1,1}$ from the first step, we obtain $L_{j,1} = A_{j,1} L_{1,1}^{-\top}$ for all $j > 1$ by solving a triangular system (on the GPU). Finally the first block column of $L$ is used to update the trailing submatrix of $A$. Note that $A_{i,j} = \sum_{k=1}^{j} L_{i,k} L_{j,k}^\top = L_{i,1} L_{j,1}^\top + \sum_{k=2}^{j} L_{i,k} L_{j,k}^\top$ for $2 \leq j \leq i$, so we can update the trailing submatrix as $A_{i,j} = A_{i,j} - L_{i,1} L_{j,1}^\top$. We implemented a parallel version of the above algorithm which distributes block-rows between the available processors in a 1D block-cyclic way (e.g. Figure 5 (left): rows 1 and 3 are assigned to GPU-1, rows 2 and 4 are assigned to GPU-2). For each column update, one processor executes the first step and transfers the result to the others (the arrows in Figure 5), which can then execute step 2 in parallel. To update the trailing matrix, further data transfer between devices may be necessary. The tile-size is chosen as a function of GPU memory: each device needs to hold one block column plus a single block at any given time. An analysis of the scalability of our implementation is in Appendix C.

### 3.3 Optimizing data transfers and other improvements.

The speed of computations on GPUs is such that data transfers to and from the devices become significant bottlenecks. We have described earlier how, for matrix-vector products, the computed blocks of $K_{nm}$ never leave the device. Further, optimization is possible by parallelizing computations and data transfers. Indeed, modern GPUs have an independent and parallel control on the following activities: loading from RAM, saving to RAM, performing computations. By running three parallel threads for the same GPU and assuming equal duration of each piece of work, we can run $t$ GPU computations in $t + 2$ time units instead of $3t$ time units for a serial implementation (see Figure 3, where $t = 3$). This guarantees near optimal usage of the GPU and in practice corresponds to a considerable speed up of matrix-vector products.

**Leveraging the trade-off numerical precision / computational power.** GPUs are designed to achieve peak performance with low precision floating point numbers, so much that going from 64 to 32-bit floats can correspond (depending on the exact architecture) to $\approx 10\times$ throughput improvement. However, changing precision can lead to unexpected problems. For example, computing the Gaussian kernel is commonly done by expanding the norm $\|\boldsymbol{x} - \boldsymbol{x}'\|^2 = \boldsymbol{x}^\top \boldsymbol{x} - 2\boldsymbol{x}^\top \boldsymbol{x}' + \boldsymbol{x}'^\top \boldsymbol{x}'$, but in high dimensions $\|\boldsymbol{x}\|, \|\boldsymbol{x}'\|$ can be very large and the cross-term very negative, so their sum has fewer significant digits. Loss of precision can lead to non positive-definite kernels causing Cholesky decomposition to fail. To avoid this, we compute $K_{mm}$ in blocks, converting each block to 64-bit precision for the sum, and then back to 32-bits.

**Dealing with thin submatrices.** As a result of our block division strategies, it may happen that blocks become thin (i.e. one dimension is small). In this case, matrix operations, e.g. using cuBLAS [33], cannot leverage the full computational power. In turn this can reduce performance, breaking the inherent computational symmetry among GPUs which is crucial for the effectiveness of a parallel system like the one proposed in this paper. To guarantee good performance for this case, instead of using standard GPU operations, we perform matrix-vector products using KeOps [8]: a specialized library to compute kernel matrices very efficiently when one dimension is small, see Table 1.

**Dealing with sparse datasets.** On the other side of the spectrum, sparse datasets with high dimensionality are common in some areas of machine learning. While the kernel computed on such datasets will be dense, and thus can be handled normally, it is inefficient and in some cases impossible (e.g. with $d \sim 10^6$ as is the case for the YELP dataset we used) to convert the inputs to a dense representation. We therefore wrapped specialized sparse linear algebra routines to perform sparse matrix multiplication [35], and adapted other operations such as the row-wise norm to sparse matrices. Thus our library handles sparse matrices with no special configuration, both on the GPU and – if a GPU is not available – on the CPU.

## 4 Large-scale experiments

We ran a series of tests to evaluate the relative importance of the computational solutions we introduced, and then performed extensive comparisons on real-world datasets. The outcome of the first tests is given in Table 1 and is discussed in Appendix A.1 for brevity. In summary, it shows a $20\times$ improvement over the base implementation of [43] which runs only partially on the GPU. Such improvement is visible in equal parts for the preconditioner computations, and for the iterative CG algorithm. For the second series of experiments we compared our implementation against three other software packages for GPU-accelerated kernel methods on several large scale datasets. All experiments were run on the same hardware, with comparable amounts of hyperparameter tuning. Finally we compared the results of our library against a comprehensive list of competing kernel methods found in the literature. We will denote our implementation by **Falkon** for squared loss and by **LogFalkon** for logistic loss. Next we present the algorithms we will compare with, then shortly describe the datasets used and the experimental setting, and finally show the benchmark results. More details are in Appendix A.

**Algorithms under test.** We compare against the following software packages: EigenPro [30], GPflow [58] and GPyTorch [16]. The first library implements a KRR solver based on preconditioned block-coordinate gradient descent where the preconditioner is based on a truncated eigendecomposition of a data subsample. EigenPro provides a fully in-core implementation and therefore does not scale to the largest datasets we tried. On some datasets EigenPro required the training data to be subsampled to avoid GPU memory issues. The other two packages implement several GP approximations and exact solvers, and we had to choose the model which would give a more appropriate comparison: we decided to avoid deep GPs [13, 63, 11] since they share more similarities

Table 1: Relative performance improvement of the implemented optimizations w.r.t. [43]. The experiment was run with the HIGGS dataset, $1 \times 10^5$ centers and 10 conjugate gradient iterations.

| Experiment | Preconditioner | | Iterations | |
|---|---|---|---|---|
| | Time | Improvement | Time | Improvement |
| Falkon from [43] | 2337 s | – | 4565 s | – |
| Float32 precision | 1306 s | 1.8× | 1496 s | 3× |
| GPU preconditioner | 179 s | 7.3× | 1344 s | 1.1× |
| 2 GPUs | 118 s | 1.5× | 693 s | 1.9× |
| KeOps | 119 s | 1× | 232 s | 3× |
| Overall improvement | | 19.7× | | 18.8× |

to deep nets than to kernel methods; on the other hand the exact GP – even when implemented on GPU [16, 59] – as well as structured kernel interpolation [62, 17] approximations do not scale to the size of datasets we are interested in. The only GP models which would scale up to tens of millions of points are stochastic variational GPs (SVGP). The SVGP is trained in minibatches by maximizing the ELBO objective with respect to the variational parameters and the model hyperparameters. Stochastic training effectively constrains GPU memory usage with the minibatch size. Hyperparameters include kernel parameters (such as the length-scale of the RBF kernel) as well as the inducing points which – unlike in Falkon – are modified throughout training using gradient descent. For this reason SVGP works well even with very few inducing points, and all operations can run in-core. While GP solvers are capable of estimating the full predictive covariance, we ensured that the software did not compute it, and further we did not consider prediction times in our benchmarks. Furthermore we always considered the Gaussian kernel with a single length-scale, due to the high effort of tuning multiple length-scales for Falkon, although for GPs tuning would have been automatic. Both GPyTorch and GPflow implement the same SVGP model, but we found the best settings on the two libraries to be different; the discrepancies in running time and accuracy between the two GP libraries come from implementation and tuning differences. We ran all algorithms under as similar conditions as possible: same hardware, consistent software versions, equal floating-point precision and equal kernels (we always considered the Gaussian kernel with a single length-scale). Hyperparameters were optimized manually by training on a small data subset, to provide a sensible trade off between performance and accuracy: we increased the *complexity* of the different algorithms until they reached high GPU utilization since this is often the knee in the time-accuracy curve. Details on the GP likelihoods, optimization details and other settings used to run and tune the algorithms are in Appendix A.4.

**Datasets.** We used eight datasets which we believe represent a broad set of possible scenarios for kernel learning in terms of data size, data type and task ranging from MSD with $5 \times 10^5$ points up to TAXI with $10^9$ points and YELP with $10^7$ sparse features. The characteristics of the datasets are shown in table 2 while a full description, along with details about preprocessing and relevant data splits, is available in appendix A.3.

**Experimental setting.** All experiments were run on a Dell PowerEdge server with 2 Intel Xeon 4116 CPUs, 2 Titan Xp GPUs and 256GB of RAM. Since out of the analyzed implementations only Falkon could use both GPUs effectively, we ran it both in a 2-GPU configuration (see Table 2) and in a single-GPU configuration (see in appendix Table 4) where Falkon was on average 1.6× slower. Each experiment was run 5 times, varying the random train/test data split and the inducing points. Out of all possible experiments, we failed to run GPyTorch on TIMIT due to difficulties in setting up a multi-class benchmark (this is not a limitation of the software). Other experiments, such as EigenPro on several larger datasets, failed due to memory errors and others yet due to software limitations in handling sparse inputs (none of the examined implementations could run the sparse YELP dataset). Finally, LogFalkon only makes sense on binary classification datasets.

**Results.** We show the results in Table 2. In all cases, our library converges in less time than the other implementations: with an average speedup ranging from 6× when compared to EigenPro to > 10× when compared to GPyTorch. Only on very few datasets such as AIRLINE-CLS, GPflow gets closer to Falkon's running time. Both models had worse accuracy than Falkon. EigenPro has generally high accuracy but can not handle large datasets at all. Finally, LogFalkon provides a small but consistent accuracy boost on binary classification problems, at the expense of higher running time. Compared with the original Falkon library [43] we report slightly higher error on HIGGS; this is attributable to

Table 2: Accuracy and running-time comparisons on large scale datasets.

| | TAXI $n \approx 10^9$ | | HIGGS $n \approx 10^7$ | | YELP $n \approx 10^6, d \approx 10^7$ | |
| | RMSE | time | $1 - $ AUC | time | rel. RMSE | time |
|---|---|---|---|---|---|---|
| Falkon | **311.7±0.1** | **3628±2 s** | 0.1804±0.0003 | **443±2 s** | **0.810±0.001** | **1008±2 s** |
| LogFalkon | — | | **0.1787±0.0002** | 2267±5 s | — | |
| EigenPro | FAIL | | FAIL | | FAIL | |
| GPyTorch | 315.0±0.2 | 37 009±42 s | 0.1997±0.0004 | 2451±13 s | FAIL | |
| GPflow | 313.2±0.1 | 30 536±63 s | 0.1884±0.0003 | 1174±2 s | FAIL | |

| | TIMIT $n \approx 10^6$ | | AIRLINE $n \approx 10^6$ | | MSD $n \approx 10^5$ | |
| | c-error | time | rel. MSE | time | rel. error | time |
|---|---|---|---|---|---|---|
| Falkon | 32.27±0.08 % | **288±3 s** | **0.758±0.005** | **245±5 s** | $(4.4834{\pm}0.0008){\times}10^{-3}$ | **62±1 s** |
| EigenPro | **31.91±0.01 %** | 1737±8 s | 0.785±0.005 | 1471±11 s[1] | **$(4.4778{\pm}0.0004){\times}10^{-3}$** | 378±8 s |
| GPyTorch | — | | 0.793±0.005 | 2069±50 s | $(4.5004{\pm}0.0010){\times}10^{-3}$ | 502±2 s |
| GPflow | 33.78±0.14 % | 2672±10 s | 0.782±0.005 | 1297±2 s | $(4.4986{\pm}0.0005){\times}10^{-3}$ | 525±5 s |

| | AIRLINE-CLS $n \approx 10^6$ | | SUSY $n \approx 10^6$ | |
| | c-error | time | c-error | time |
|---|---|---|---|---|
| Falkon | 31.5±0.2 % | **186±1 s** | 19.67±0.02 % | **22±0 s** |
| LogFalkon | **31.3±0.2 %** | 1291±3 s | **19.58±0.03 %** | 83±1 s |
| EigenPro | 32.5±0.2 % | 1629±1 s[1] | 20.08±0.55 % | 90±0 s[2] |
| GPyTorch | 32.5±0.2 % | 1436±2 s | 19.69±0.03 % | 882±9 s |
| GPflow | 32.3±0.2 % | 1039±1 s | 19.65±0.03 % | 560±11 s |

[1]Using a random subset of $1{\times}10^6$ points for training. [2]Using a random subset of $6{\times}10^5$ points for training.

the use of low-precision floating point numbers. We did not find significant performance differences for other datasets. We defer comparisons with results from the literature to Appendix A.6; suffice it to note that a distributed GP applied to the TAXI dataset resulted in a running-time of 6000 s using a system with 28 000 CPUs [37] while we achieved similar accuracy in less time, with a much smaller computational budget.

## 5    Conclusions

Making flexible and easy to use machine learning libraries available is one of the keys of the recent success of machine learning. Here, we contribute to this effort by developing a library for large scale kernel methods. We translate algorithmic ideas into practical solutions, using a number of carefully design computational approaches specifically adapted to the GPU. The resulting library achieves excellent performance both in terms of accuracy and computational costs. A number of further developments are possible building on our work. For example, considering other loss functions or optimization approaches, and especially more structured kernels [9] that could further improve efficiency.

## Broader Impact

This work has the potential to greatly speed up a certain class of machine learning workloads, namely kernel methods on large datasets when GPU(s) are available. If deployed widely, the positive impact of the presented method could be twofold: on the one hand it may reduce electricity consumption necessary to run such large-scale predictions [15], thus positively impacting the environment; on the other hand it could enable analysis of large datasets which were previously only accessible to simpler methods. At the same time the speedup we obtain relies on GPU accelerators; since this type of hardware is generally expensive, it could increase disparity in access to fast algorithms.

## Acknowledgments

This material is based upon work supported by the Center for Brains, Minds and Machines (CBMM), funded by NSF STC award CCF-1231216, and the Italian Institute of Technology. We gratefully acknowledge the support of NVIDIA Corporation for the donation of the Titan Xp GPUs and the Tesla k40 GPU used for this research. Part of this work has been carried out at the Machine Learning Genoa (MaLGa) center, Università di Genova (IT) L. R. acknowledges the financial support of the European Research Council (grant SLING 819789), the AFOSR projects FA9550-17-1-0390 and BAA-AFRL-AFOSR-2016-0007 (European Office of Aerospace Research and Development), and the EU H2020-MSCA-RISE project NoMADS - DLV-777826. This work was funded in part by

the French government under management of Agence Nationale de la Recherche as part of the "Investissements d'avenir" program, reference ANR-19-P3IA-0001 (PRAIRIE 3IA Institute).

## Footnotes

[1] https://github.com/FalkonML/falkon

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
