[Supplementary Material]

# A Further experiment details and results

## A.1 Relative impact of performance optimizations

We performed an experiment to analyze how much improvement was due to the different performance optimization steps. We ran Falkon on the HIGGS dataset several times with the same hyperparameters ($m = 1 \times 10^5$ and 10 epochs), but with different *features* enabled. Each feature roughly corresponds to one of the performance optimizations discussed in Section 3. Our baseline model is very similar to the original Falkon implementation [43], where the preconditioner ran on the CPU, float64 precision was being used, but matrix-vector multiplications for the CG algorithm were GPU accelerated. As a first optimization we used float32 precision for all computations, with care taken to avoid errors in the Cholesky decomposition as discussed in Section 3. This immediately resulted in a $2\times$ speedup for the CPU part, and $3\times$ for the GPU part. Switching to a GPU preconditioner (using the algorithms described in Appendix C) gave a huge boost to the preconditioner running time which went from more than $20\,\mathrm{min}$ to just under $3\,\mathrm{min}$. Adding a second GPU produced a perfect $2\times$ speedup for the CG iterations, and a more modest $1.5\times$ speedup for the preconditioner which a) involves operations which are not perfectly parallelizable and b) incurs in some fixed startup costs. Finally, since the HIGGS dataset has only 9 features (thus the data matrix is thin), we can use KeOps [8] with great benefits to the speed of matrix-vector multiplications. Overall our implementation provides a nearly $20\times$ improvement over the baseline, which makes learning on several huge datasets doable in a matter of minutes.

## A.2 Multi-GPU scalability

In this section we look into the scalability of our implementation across multiple GPUs. Scalability results for the full Falkon algorithm on the TAXI dataset are shown in Figure 6. This result depends on scaling both the preconditioner and the conjugate gradient iterations. The preconditioner itself is computed with three main operations: two Cholesky decompositions and one triangular matrix multiplication (this is called the LAUUM operation in LAPACK terms), see Figure 4 for more details. Each CG iteration instead consists of two multiplications between the kernel matrix and an arbitrary vector. First we look at the scalability of the preconditioner operations with multiple GPUs. Then we examine our out-of-core matrix-vector product implementation and compare it to KeOps for different settings of $n$ and $d$.

Figure 6: Multi-GPU scalability of Falkon on the TAXI dataset (settings are the same as per Table 3). Falkon scales remarkably well, with even 4 GPUs.

**Preconditioner scalability.** Figure 7 shows the results from running both triangular matrix multiplication and the Cholesky decomposition with one and two GPUs. At low matrix sizes the speedup with two GPUs is negligible, especially for the Cholesky decomposition. In such cases it is best to use a single GPU (especially since for $n = 40000$ the whole matrix fits in GPU memory, so an in-place decomposition can be used). With higher matrix sizes, having more than one GPU starts bringing real benefits, with a peak speedup around $1.8\times$ for preconditioners of size $140\,000$. The

(a) Parallel LAUUM.

(b) Parallel Cholesky decomposition.

Figure 7: Running time of two preconditioner operations with one and two GPUs. The relative speed-up with 2 GPUs is shown in the black dashed line. The LAUUM operation (triangular matrix multiplication) was run out-of-place, which is theoretically easier to parallelize, while the Cholesky decomposition was run in-place.

factors blocking such speedup from reaching a perfect $2\times$ are different for the two operations. Since the LAUUM operation was run out-of-place (see Appendix C for more details), it does not need any synchronization – and should therefore be able to scale well across multiple GPUs. The main blocking factor is the operation at Line 7 of Algorithm 3 which is executed on the CPU (since an equivalent implementation does not exist in cuSOLVER), thus both GPU threads must share the same CPU resources. We left porting the LAUUM operation to the GPU as future work, but it has the potential to speed up the LAUUM operation considerably. For the Cholesky decomposition the limiting factors are the data-dependencies intrinsic to the algorithm which cannot be easily solved.

**Comparing different MVM implementations.** We compare our specialized routine for the kernel-vector multiplication $k(X^{(1)}, X^{(2)})v$ implemented in Python, leveraging PyTorch for GPU computations, against the native CUDA implementation from KeOps [8]. Using a similar notation for the dimensions as in the main text we have $X^{(1)} \in \mathbb{R}^{n \times d}, X^{(2)} \in \mathbb{R}^{m \times d}, v \in \mathbb{R}^{m \times 1}$ and $k(\cdot, \cdot)$ is a kernel function. Two distinct scenarios arise in different settings: increasing the number of data points $n$ produces linear scaling for both implementations, with KeOps being approximately 10 times faster than our implementation (see Figure 8(a)). Increasing the data dimensionality $d$ our implementation scales linearly, but KeOps scales polynomially, so as it is obvious from Figure 8(b) KeOps can not be used when the data is high-dimensional. A caveat of this plot is that KeOps is continuously evolving, and is likely to improve performance with large $d$ in the future. In our final algorithm we set a threshold on the data dimensionality and switch implementation based on this. Finally note that this operation scales almost perfectly with multiple GPUs.

### A.3 Additional information on the datasets

We used several datasets which we believe represent a broad set of scenarios for kernel learning, in terms of data size, data type, and learning task. We normally used a standard random split with 80% training, 20% testing data unless predefined splits existed (as noted below). Preprocessing mostly consisted in basic data cleaning and data standardization to zero mean and unit standard deviation; we comment in more detail below on specific preprocessing steps applied to the individual datasets.

**HIGGS** has dimensions $n = 1.1 \times 10^7, d = 28$ and a binary target. It was preprocessed to 0 mean and unit variance. Results are reported on a 80-20 split with 1 minus the AUC metric in Table 2 and with the binary classification error in Table 6. It is available for download at `https://archive.ics.uci.edu/ml/datasets/HIGGS`.

**TIMIT** has dimensions $n = 1.2 \times 10^6, d = 440$ and a multiclass target with 144 classes. TIMIT comes from audio data, and our dataset uses the $10\,\mathrm{ms}$ resampling rate as in [29, 30]. It was preprocessed to 0 mean and unit standard deviation. The error metric is classification error on a

Figure 8: Scaling of matrix-vector implementations where the matrix is the Gaussian kernel. In (a) we have set $m = 20\,000$, $d = 10$ and $n$ is variable; in (b) we set $m = n = 20\,000$ and we vary $d$. All experiments are run on 1 and 2 GPUs on single precision random data.

subset of classes (as used in [29]), and is calculated over a standardized subset of $57\,242$ samples. It is available for download at `https://catalog.ldc.upenn.edu/LDC93S1`.

**YELP** has dimensions $n = 1.5 \times 10^6$, $d = 6.52 \times 10^7$ and a continuous target. This dataset consists of text reviews, labeled with their star rating. We used the same data as [57] (Yelp round 9 dataset), processed by extracting all 3-grams and encoding each review by a count vector which tells us which 3-grams are present. Such encoding produces a large number of sparse features which is reflected in the huge dimensionality of this dataset. Since the data is sparse we did not normalize it. The error metric is RMSE, calculated on random 20% of the samples. The dataset can be provided on request.

**TAXI** has dimensions $n = 1.1 \times 10^9$, $d = 9$ with a continuous target. Data are normalized to have zero-mean and unit standard deviation; reported error is RMSE on a 20% random sub-sample. The data can be downloaded by following instructions at `https://github.com/toddwschneider/nyc-taxi-data`. Consistently with other users of this dataset [37] we took the data from January 2009 to December 2015, excluding outliers (taxi trips more than 5 hours long) and trips where the pickup or drop off location is outside of NYC.

**AIRLINE** has dimensions $n = 5.93 \times 10^6$, $d = 8$ and a continuous target. Data are normalized to zero-mean and unit standard deviation, and the error is the MSE over normalized targets calculated on random test-sets of size $33\,\%$ of the full data (consistently with the literature [21, 19]). The same dataset is also used for binary classification by thresholding the target at 0, which results in the **AIRLINE-CLS** dataset. For this latter variation we used $100\,000$ random points for testing, reporting classification error in Table 2 and 1 minus the AUC in Table 6 to facilitate comparisons with the literature. The data can be downloaded from `https://www.transtats.bts.gov/Fields.asp?Table_ID=236` and `http://stat-computing.org/dataexpo/2009/supplemental-data.html`.

**MSD** has dimensions $n = 5.1 \times 10^5$, $d = 90$ with continuous target. Data are normalized to zero-mean and unit standard deviation, and we report the relative error over a standard test-set of size $51\,630$. The dataset can be downloaded from `https://archive.ics.uci.edu/ml/datasets/YearPredictionMSD`.

**SUSY** has dimensions $n = 5 \times 10^6$, $d = 18$ with binary target. Data are normalized to zero-mean and unit standard deviation. We report the classification error on 20% of the data. Data is available from the UCI repositories `https://archive.ics.uci.edu/ml/datasets/SUSY`.

### A.4 Additional information on the experimental settings

1. EigenPro2. Its only hyperparameters – other than the kernel parameters – are the ones governing the preconditioner's complexity making EigenPro easy to tune. It is however

limited to datasets which fit entirely in GPU memory, so can not easily scale to larger datasets; to alleviate this problem, consistently with the original paper, some experiments were run on sub-sampled datasets. Furthermore, on some experiments we found it necessary to manually tune the learning rate (we divided the automatically inferred learning rate by a fixed integer, denoted by $\eta\div$ in Table 3).

2. GPFlow (v2.1.3). We used the SVGP model with Gaussian likelihood for regression, Bernoulli for binary classification and Softmax for multi-class problems. We used Adam for optimization and tuned the learning rate, the number of inducing points, and the constraints on the variational distribution covariance (i.e. diagonal or full covariance matrix). We found that using a full covariance matrix was rarely beneficial and increased training times slightly, so all final experiments used a diagonal covariance matrix. The number of parameters was $m \times d + m \times 2 + 3$ which includes the inducing points, the variational parameters, two parameters for the Gaussian kernel (lengthscale and variance) and the variance of the likelihood. For multi-class problems separate variational parameters were trained for each class. Since we wished to use single-precision floating point numbers in order to make GPU training more efficient, we found that natural gradient optimization was unstable. It remains to be seen whether the tradeoff between double-precision data and natural gradient optimization could further improve results. We further tested the benefits of using whitening of the inducing points, and found that it decreased per-epoch running times by about $2\times$, while at the same time slowing down convergence by around the same amount. In practice this meant that the difference in global running time was not strongly affected by whitening, and we ended up using it only for the HIGGS data.

3. GPyTorch (v.1.2.0). We used the SVGP model with Gaussian and Bernoulli likelihoods. We were unable to run GPyTorch's SVGP model on the TIMIT dataset due to problems in dealing with multiple outputs. We used the natural gradient optimizer to learn the variational parameters, and Adam to learn the other hyperparameters. The learning rate of the two optimizers was kept equal and tuned for best performance. We further optimized the number of inducing points, and variational distribution constraints. In practice we found that we had to use the natural gradient variational distribution for regression problem, and the lower-triangular parametrization for classification problems (which are non-conjugate). We additionally tested whether whitening the inducing points was beneficial: in practice we found that using the unwhitened strategy was around $3\times$ faster and did not hamper convergence, so we selected it for all experiments. While GPyTorch is theoretically able to run on multi-GPU systems, we noticed that this feature was not available for the SVGP model thus we always used a single GPU; furthermore, while a KeOps integration into GPyTorch is available, we found that for the SVGP model it would increase the running time, so we did not use it. The trained parameters were the same as for GPFlow plus another scalar for the GP mean.

4. Falkon. We tuned the kernel length-scale, number of inducing points and regularization amount. We used a coarse to fine approach to tune the length-scale which gives good results with a limited number of validation runs.

5. Logistic Falkon. Here we tuned the kernel length-scale, number of inducing points and regularization path. We found that the algorithm is not very sensitive to the exact regularization path: it is sufficient to set the final $\lambda$, and many different paths which lead to such value will work in the same way.

## A.5 Additional benchmarks

In Table 4 we show the performance of the Falkon algorithm on all considered datasets for 1 and 2 GPUs side by side. It is clear that larger datasets scale better with more GPUs since the startup cost (mostly taken up by CUDA initialization) and the lower scaling ratio of the preconditioner are amortized.

In Table 5 we compare the running times of Falkon and ThunderSVM [60] on three popular image datasets. ThunderSVM was chosen among several SVM implementations as it runs entirely on the GPU, and can thus solve the hinge-loss problem quickly for problems of moderate size. Smaller datasets than the ones used for previous experiments were considered, since ThunderSVM solves the full SVM problem and thus suffers from cubic time scaling. The results obtained show that

Table 3: Summary of the most important hyperparameter settings for all algorithm-dataset combinations. We denote by $\eta$ the learning rate, by *subsample* the amount of training-set subsampling that was performed (i.e. training was done on a smaller dataset), and by Newton steps the number of separate runs of the main Falkon algorithm for Logistic Falkon (see Appendix B).

| | | AIRLINE | AIRLINE-CLS | MSD | SUSY | TIMIT | YELP | HIGGS | TAXI |
|---|---|---|---|---|---|---|---|---|---|
| | n | $5.93\times10^6$ | $5.93\times10^6$ | $5.1\times10^5$ | $5\times10^6$ | $1.2\times10^6$ | $1.6\times10^6$ | $11\times10^7$ | $1.15\times10^9$ |
| | d | 8 | 8 | 90 | 18 | 440 | $6.5\times10^7$ | 28 | 9 |
| | labels | reg | 2-cls | reg | 2-cls | 144-cls | reg | 2-cls | reg |
| Falkon | m | $1\times10^5$ | $1\times10^5$ | $5\times10^4$ | $3\times10^4$ | $1\times10^5$ | $5\times10^4$ | $1.2\times10^5$ | $1\times10^5$ |
| | $\sigma$ | 0.9 | 0.9 | 7 | 3 | 14.5 | 20 | 3.8 | 1 |
| | $\lambda$ | $1\times10^{-8}$ | $1\times10^{-8}$ | $2\times10^{-6}$ | $1\times10^{-6}$ | $5\times10^{-9}$ | $1\times10^{-6}$ | $3\times10^{-8}$ | $2\times10^{-7}$ |
| | epochs | 20 | 10 | 10 | 5 | 5 | 10 | 10 | 7 |
| LogFalkon | m | – | $1\times10^5$ | – | $2\times10^4$ | – | – | $1\times10^5$ | – |
| | $\sigma$ | – | 0.9 | – | 3 | – | – | 5 | – |
| | $\lambda$ | – | $1\times10^{-9}$ | – | $1\times10^{-8}$ | – | – | $1\times10^{-9}$ | – |
| | Newt. steps | – | 9 | – | 6 | – | – | 9 | – |
| GPyTorch | m | 2000 | 2000 | 3000 | 2000 | – | – | 2000 | 1000 |
| | $\eta$ | $5\times10^{-3}$ | $2\times10^{-3}$ | $2\times10^{-3}$ | $1\times10^{-3}$ | – | – | $2\times10^{-2}$ | $2\times10^{-3}$ |
| | epochs | 20 | 20 | 20 | 20 | – | – | 15 | 5 |
| GPflow | m | 2000 | 2000 | 3000 | 2000 | 2000 | – | 2000 | 1000 |
| | $\eta$ | $5\times10^{-3}$ | $5\times10^{-3}$ | $2\times10^{-3}$ | $3\times10^{-3}$ | $1\times10^{-2}$ | – | $2\times10^{-2}$ | $3\times10^{-3}$ |
| | epochs | 25 | 20 | 45 | 10 | 15 | – | 60 | 10 |
| | whiten | no | no | no | no | no | – | yes | no |
| EigenPro | $\eta\div$ | 10 | 12 | 20 | 1 | 1 | – | – | – |
| | subsample | $1\times10^6$ | $1\times10^6$ | – | $6\times10^5$ | – | – | – | – |
| | epochs | 9 | 10 | 9 | 1 | 4 | – | – | – |

Table 4: Benchmark timings using a single GPU. The relative slowdown with respect to Falkon on 2 GPUs is also provided for comparison with Table 2.

| | 1 GPU | 2 GPUs | Relative change |
|---|---|---|---|
| TAXI | $7215\pm4$ s | $3628\pm2$ s | $1.99\times$ |
| HIGGS | $715\pm6$ s | $443\pm2$ s | $1.61\times$ |
| YELP | $1981\pm6$ s | $1008\pm2$ s | $1.97\times$ |
| TIMIT | $416\pm4$ s | $288\pm3$ s | $1.44\times$ |
| AIRLINE | $334\pm2$ s | $245\pm5$ s | $1.36\times$ |
| MSD | $81\pm0$ s | $62\pm1$ s | $1.31\times$ |
| AIRLINE-CLS | $391\pm5$ s | $269\pm3$ s | $1.45\times$ |
| SUSY | $29\pm1$ s | $22\pm0$ s | $1.32\times$ |

Falkon can work efficiently even on smaller datasets, resulting between 2 and 10 times faster than ThunderSVM (depending on problem size), with comparable accuracy. To further shave off some time, we implemented a version of Falkon which runs entirely inside the GPU: we call this **InCoreFalkon**, and it can only be used on smaller datasets which fit inside the GPU, leaving some space to spare which is used for the preconditioner and the other computations. Table 5 shows that InCoreFalkon gives a further speed-up of – on average – $2\times$ compared to the standard implementation.

Table 5: Comparing the running times of Falkon, the in-core version of Falkon and ThunderSVM on three image datasets. Hyperparameters (especially the number of inducing points $m$) were tuned so that the two algorithms obtained approximately the same accuracy.

| | MNIST $n = 6\cdot10^4, d = 780$ | CIFAR10 $n = 6\cdot10^4, d = 1024$ | SVHN $n = 7\cdot10^4, d = 1024$ |
|---|---|---|---|
| Falkon | 10.9 s | 13.7 s | 17.2 s |
| InCoreFalkon | 6.5 s | 7.9 s | 6.7 s |
| ThunderSVM | 19.6 s | 82.9 s | 166.4 s |

Table 6: Survey of results on the datasets we considered, as reported in the literature. We report the result of our implementation (Falkon) next to other implementations, along with the time taken and the hardware used (where available).

| Dataset | Falkon | | Other methods | | |
|---|---|---|---|---|---|
| | error | time | error | time | reference |
| TAXI (metric: RMSE) | 311.7±0.1 | 3628±2 s | 309.7 | 6000 s 28 000 vCPUs (AWS) | ADVGP [37] |
| HIGGS (metric: c-err) | 25.78±0.03 % | 443±2 s | 32.87 % | 1392 s on 14 node cluster | liquidSVM [51] |
| YELP (metric: RMSE) | 0.810±0.001 | 1008±2 s | 0.861 | ≈ 3500 s | Nyström [57] |
| | | | 0.854 | ≈ 30 000 s on 128 machines (AWS) | Full linear kernel [57] |
| AIRLINE (metric: MSE) | 0.758±0.005 | 245±5 s | 0.827±0.004 | 265±6 s on a laptop | VFF-GP [21] |
| | | | 0.791±0.005 | 18 360±360 s on a cluster | SVIGP [21] |
| MSD (metric: rel. err.) | $4.48\times10^{-3}$ | 62±1 s | $\approx 4.55\times10^{-3}$ | 210 s on IBM POWER8 | Hierarchical [9] |
| | | | $4.58\times10^{-3}$ | 289 s on 8 r3.8xlarge (AWS) | Faster KRR [3] |
| AIRLINE-CLS (metric: AUC) | 0.739±0.002 | **186±1 s** | 0.781±0.001 | 14 328 s | Varitional Deep GP [63] |
| | | | 0.694 | 5200 s | TT-GP [22] |
| | | | 0.788 | 1375 s | Deep TT-GP [22] |
| | | | 0.665 | 80 000 s | cVGP[20] |
| | | | 0.785 | ≈ 5000 s | RF Deep GPs [11] |
| SUSY (metric: c-err) | 19.67±0.02 % | **22±0 s** | ≈ 20% | ≈ 2000 s on IBM POWER8 | Hierarchical [9] |
| | | | 19.8% | 58 s on 1 Titan Xp | EigenPro 2.0 [30] |

## A.6 Performance comparisons in a literature review

We scanned the literature for results which used kernel methods on the datasets considered in this paper, which reported both accuracy and running times. This allowed us to confirm that the results reported in our benchmarks (see Table 2) were in-line with what had been previously reported. The outcome is shown in Table 6. We do not report results where running time is not mentioned. Some of the numbers in Table 6 have higher accuracy than Falkon: this comes from the use of deep GPs which – through a vast number of parameters – can learn better data representations. Such models are intrinsically different in spirit from kernel methods, and we do not aim to compare with them specifically; they are reported in Table 6 for the sake of completeness.

## B  Logistic Falkon Algorithm

In this section we provide some more details on how to derive fast algorithms with strong theoretical guarantees for smooth loss functions beyond squared loss. In particular, the main ideas from a theoretical and algorithmic viewpoint that we are going to recall here are developed in [32], [31]. Our goal, as stated in the main text, is to make these ideas practical, by efficiently implementing and deploying the algorithms and making full use of the available computational architectures. In particular, we will focus on the following set of *generalized self concordant* loss functions:

**Definition 1.** Generalized self-concordant (GSC) function [32] *Let $\mathcal{H}$ be a Hilbert space and let $z = (x, y)$ be an input-output pair. We say that $\ell_z : \mathcal{H} \to \mathbb{R}$ is a generalized self-concordant function on $\mathcal{G} \subset \mathcal{H}$, when $\mathcal{G}$ is a bounded subset of $\mathcal{H}$ and $\ell_z$ is a convex and three times differentiable*

---

**Algorithm 2** Pseudocode for appr. Newton method with Falkon, for GSC losses (based on [31]).

---

1: **function** GSC-FALKON($X \in \mathbb{R}^{n \times d}, \boldsymbol{y} \in \mathbb{R}^n, \lambda, m, t, T$)
2:     Set $\alpha_0 = 0 \in \mathbb{R}^m$ and $\mu_0 > 0, q > 0$ according to [31].
3:     $X_m, \boldsymbol{y}_m \leftarrow$ RANDOMSUBSAMPLE($X, \boldsymbol{y}, m$)
4:     **for** $k \in \mathbb{N}$ **do**
5:         $f_{k+1} \leftarrow$ WEIGHTEDFALKON($X, \boldsymbol{y}, X_m, \boldsymbol{y}_m \mu_k, t, \alpha_k$)
6:         $\mu_{k+1} \leftarrow q\mu_k$
7:         Stop when $\mu_{k+1} < \lambda$ and set $\alpha_{last} \leftarrow \alpha_k$.
8:     **end forreturn** $\widehat{\alpha} \leftarrow$ WEIGHTEDFALKON($X, \boldsymbol{y}, X_m, \boldsymbol{y}_m, \lambda, T, \alpha_k$)
9: **end function**

<br>

1: **function** WEIGHTEDFALKON($X \in \mathbb{R}^{n \times d}, \boldsymbol{y} \in \mathbb{R}^n, X_m \in \mathbb{R}^{m \times d}, \boldsymbol{y}_m \in \mathbb{R}^m, \lambda, t, \alpha_0 \in \mathbb{R}^m$)
2:     $T, A \leftarrow$ WEIGHTEDPRECONDITIONER($X_m, \boldsymbol{y}_m, \alpha_0, \lambda$)
3:     **function** LINOP($\boldsymbol{\beta} \in \mathbb{R}^m$)
4:         $\boldsymbol{v} \leftarrow A^{-1}\boldsymbol{\beta}$
5:         $z \leftarrow k(X, X_m)\boldsymbol{\beta}$                                          ▷ predictions on the dataset
6:         $D \leftarrow \texttt{diag}[(\ell^{(2)}((z)_1, (\boldsymbol{y})_1), \ldots, \ell^{(2)}((z)_n, (\boldsymbol{y})_n))]$
7:         $\boldsymbol{c} \leftarrow k(X_m, X)Dk(X, X_m)T^{-1}\boldsymbol{v}$
8:         **return** $A^{-\top}T^{-\top}\boldsymbol{c} + \lambda n\boldsymbol{v}$
9:     **end function**
10:     $R \leftarrow A^{-\top}T^{-\top}k(X, X_m)\boldsymbol{y}$
11:     $\boldsymbol{\beta} \leftarrow$ CONJUGATEGRADIENT(LINOP, $R, t, \alpha_0$)                ▷ CG solver starting from $\alpha_0$
12:     **return** $T^{-1}A^{-1}\boldsymbol{\beta}$
13: **end function**

<br>

1: **function** WEIGHTEDPRECONDITIONER($X_m \in \mathbb{R}^{m \times d}, \boldsymbol{y}_m \in \mathbb{R}^m, \alpha \in \mathbb{R}^m, \lambda$)
2:     $K_{mm} \leftarrow k(X_m, X_m)$                               ▷ Compute the kernel between inducing points
3:     $z \leftarrow K_{mm}\alpha$                                              ▷ predictions on the Nyström points
4:     $T \leftarrow \text{chol}(K_{mm})$
5:     $D \leftarrow \texttt{diag}[(\ell^{(2)}((z)_1, (\boldsymbol{y}_m)_1), \ldots, \ell^{(2)}((z)_m, (\boldsymbol{y}_m)_m))]$
6:     $K_{mm} \leftarrow 1/m TDT^{\top} + \lambda \boldsymbol{I}$
7:     $A \leftarrow \text{chol}(K_{mm})$
8:     **return** $T, A$
9: **end function**

---

Note: LINOP performs the multiplications via matrix-vector products.

---

*mapping on $\mathcal{H}$ such that*

$$\forall f \in \mathcal{H}, \ \forall h, k \in \mathcal{H}, \ \nabla^{(3)}\ell_z(f)[h, k, k] \leq \sup_{g \in \mathcal{G}} |g \cdot h| \ \nabla^2 \ell_z(f)[k, k].$$

Denote by $R$ the quantity $\sup_{g \in \mathcal{G}} \|g\| < \infty$. For many loss functions $\mathcal{G}$ is just the ball in $\mathcal{H}$ centered in zero and with radius $R > 0$, then $\sup_{g \in \mathcal{G}} |g \cdot h| = R\|h\|)$. The following loss functions, which are widely used in machine learning, are generalized self-concordant

**Example 1.** (Application to finite-sum minimization [32]) *The following loss functions are generalized self-concordant functions:*
*(a) Logistic regression:* $\ell_z(f) = \log(1 + \exp(-yf(x)))$, *where* $z = (x, y)$ *with* $x \in X$ *and* $y \in \{-1, 1\}$.
*(b) Robust regression:* $\ell_z(f) = \varphi(f(x) - y)$ *with* $\varphi(u) = \log(e^u + e^{-u})$. *Here* $z = (x, y)$ *with* $x \in X$ *and* $y \in \mathbb{R}$
*(c) Softmax regression:* $\ell_z(f) = \log(\sum_{j=1}^{k}[f(x)]_j) - [f(x)]_y$, *where now* $f : X \to \mathbb{R}^k$, $z = (x, y)$, *with* $y \in \{1, \ldots, k\}$ *and* $v_j$ *denotes the j-th column of* $v \in \mathbb{R}^k$.
*(d) generalized linear models with bounded features, which include conditional random fields (see more details in [32]).* Note, in particular, that the loss functions above are generalized self concordant, but not *self concordant* as discussed in [32].

For the learning problem in Eq. (1) with generalized self-concordant loss functions, a strong theoretical result analogous to the one for kernel ridge regression (3) has been obtained [32]. In particular, the regularized empirical risk minimization solution (2) with generalized self-concordant losses achieves the bound

$$L(\hat{f}_\lambda) - \inf_{f \in \mathcal{H}} L(f) = \mathcal{O}\left(n^{-1/2}\right), \tag{10}$$

under standard regularity conditions on the learning problem and achieves fast learning rates similar to kernel ridge regression, considering more refined regularity conditions that are a natural extension of the conditions for kernel ridge regression [32].

The paper [31] suggests to solve the regularized empirical risk minimization problem (2) for generalized self-concordant losses, by using a set of techniques that are extensions of the Falkon algorithm in [43]. In particular, the problem is cast in terms of an approximate Newton method, with pseudocode shown in function GSC-Falkon of Algorithm 2. Nyström method is used a first time to reduce the size of the problem, and then a second time to derive an approximate Newton step [31]. Indeed a model of the form (4) is considered and the preconditioner now plays the role of approximate Hessian, to perform the iterated approximation Newton. Given $(\tilde{x}_j, \tilde{y}_j)_{j=1}^m$ selected uniformly at random from the dataset, the approximate Hessian $\tilde{H}$ at the step $k$ is a weighted version of the Falkon preconditioner and has the form

$$\tilde{H} = \frac{1}{m} T \tilde{D}_k T^\top + \mu_k I,$$

where $T$ is such that $T^\top T = K_{mm}$ (e.g. it is the Cholesky decomposition of $K_{mm}$) and $\tilde{D}_k \in \mathbb{R}^{m \times m}$ is a diagonal matrix whose $j$th element is $\ell^{(2)}(f_k(\tilde{x}_j), \tilde{y}_j)$ where we assume that the loss function is $\ell(f(x), y)$ and the second derivative is taken with respect to the first variable. As for Falkon, the approximate Hessian is never built explicitly, we compute instead its Cholesky decomposition in terms of the matrices $T, A$ as $\tilde{H}^{-1} = \tilde{P}\tilde{P}^\top$ with $\tilde{P} = T^{-1}A^{-1}$, see the function WeightedPreconditioner in Alg. 2. Then conjugate gradient is applied to the preconditioned problem, to solve the equation

$$\tilde{P}^\top(K_{nm}^\top D_k K_{nm} + \lambda I)\tilde{P}\beta = \tilde{P}^\top K_{nm}^\top g_k.$$

where $D_k \in \mathbb{R}^{n \times n}$ is a diagonal matrix whose $i$th element is $\ell^{(2)}(f_k(x_i), y_i)$ and $g_k \in \mathbb{R}^n$ corresponds to $(g_k)_i = \ell^{(1)}(f_k(x_i), y_i)$. To conclude, as proven in [31], to achieve the same learning rate of (10) and good practical performances, GSC-Falkon (Alg. 2) needs to call WeightedFalkon only a small number of times with decreasing regularization parameters. Moreover, each time WeightedFalkon needs to execute only few iterations of the CG algorithm. The algorithm presented in Alg. 2 has an important theoretical appeal appealing as proved in [31] since it is the fastest to date to achieve optimal learning rates for generalized self-concordant loss functions. The goal of our work is to make it also appealing from a practical viewpoint. This requires efficiently implementing and deploying Alg. 2, making full use of the available computational architectures. Clearly the main bottlenecks here are the same of Falkon for squared loss and they are introduced and discussed in Section 3.

## C    Out-Of-Core Algorithms

In this section we describe more in detail the out-of-GPU core algorithms for 1) Cholesky decomposition of a positive definite matrix and 2) multiplication of a triangular matrix by its transpose. Both algorithms use a similar technique of dividing the input matrix in smaller tiles such that operations can be performed in-core on the individual tiles. Then the main challenges of such algorithms consist in choosing when to bring which tiles in-core, and how to do so in parallel, handling data-dependencies between different tiles.

We handle parallelism between multiple GPUs using a static work-allocation scheme where the input matrix is divided into block rows or columns (made up of several tiles), and each GPU is assigned one or more such rows (or columns) block-cyclically, to ensure that the workload is approximately balanced. Ensuring a balanced workload is tricky since the input matrices are triangular, and for example a row at the top of a lower-triangular matrix will have many more tiles than a row towards the bottom of said matrix. Smaller tile-sizes (so thinner block rows/columns) make each processor's workload more even, but – in case the input matrix is not big enough – they reduce overall GPU utilization.

---

**Algorithm 3** Out-of-core LAUUM operation on an upper-triangular matrix. The algorithm's inputs are matrix $U$, a synchronization object `barrier`, an array of arrays describing which row indices are assigned to which processor `blockAllocs`, and the number of tiles per side $N$. The function described below should be called for every available GPU with a different `procId` value.

1: **function** OOCLAUUM($U \in \mathbb{R}^{n \times n}$, `barrier`, `blockAllocs`, `procId`, $N$)
2:     **for** $i = 1, \ldots, N$ **do**
3:         $C \in \mathbb{R}^{t \times t \cdot (N-i)} \leftarrow$ `ToGPU`$\left(\text{procId}, \left[U_{i,i}, \ldots, U_{i,N}\right]\right)$
4:         `barrier.wait()`
5:         **for** $j \in$ `blockAllocs`$[\text{procId}]$ **do**
6:             **if** $i = j$ **then**
7:                 $C_1 \leftarrow C_1 C_1^\top$                                          ▷ via LAUUM
8:                 **if** $i \neq N$ **then**
9:                     $C_1 \leftarrow C_1 + C_{1:(N-i+1)} C_{1:(N-i+1)}^\top$           ▷ via SYRK
10:                 **end if**
11:             **else if** $j > i$ **then**
12:                 $D \in \mathbb{R}^{t \times t \cdot (N-j)} \leftarrow$ `ToGPU`$\left(\text{procId}, \left[U_{j,j}, \ldots, U_{j,N}\right]\right)$
13:                 $C_{(j-i)} \leftarrow C_{(j-i)} D_1^\top$                                  ▷ via TRMM
14:                 **if** $j \neq N$ **then**
15:                     $C_{(j-i)} \leftarrow C_{(j-i+1):(N-i+1)} D_{2:(N-j+1)}^\top$      ▷ via GEMM
16:                 **end if**
17:             **end if**
18:             $U_{i,j} \leftarrow$ `FromGPU`$(\text{procId}, C_{(j-i)})$
19:         **end for**
20:     **end for** **return** $U$
21: **end function**

---

**Triangular matrix multiplication.** We begin by describing OOC triangular matrix multiplication, an operation which is known as LAUUM within the LAPACK library. Given an input upper triangular matrix $U \in \mathbb{R}^{n \times n}$, we want to calculate the upper triangle of $UU^\top$ and store it in the upper part of $U$ (thus making this an in-place operation). We divide $U$ in $N \times N$ tiles of size $t$ (uneven tile sizes are possible, and indeed necessary to support all input sizes, but omitted from the description for clarity), and we index all matrices by their tiles: $U_{2,2}$ is the square tile at the second block-row and second block-column of $U$. The in-place LAUUM operation can be compactly described as $U_{i,j} = \sum_{k=j}^{N-1} U_{i,k} U_{j,k}^\top$ for $j \geq i$: to update a tile of $U$ we need to multiply two block-rows of the original matrix. However, we can exploit the triangular structure of some of the above matrix multiplications to improve performance: for example, when $i = j$ it is possible to split the update into two parts $U_{i,i} = U_{i,i} U_{i,i}^\top + \sum_{k=i}^{N} U_{i,k} U_{i,k}^\top$ where the first part consists of an in-core LAUUM operation and the second of a symmetric matrix multiplication (BLAS routine SYRK) which can be up to twice as fast as the general matrix multiplication routine. Similarly, for $i < j$, $U_{i,j} = U_{i,j} U_{j,j}^\top + \sum_{k=j+1}^{N} U_{i,k} U_{j,k}^\top$ where the first part can use the TRMM routine from the BLAS library and the second must use the generic GEMM routine. To avoid overwriting parts of $U$ which are still needed for the updates – especially in a multi-GPU setting – the rows of $U$ are to be updated one at a time, from top to bottom. To ensure synchronization between multiple GPUs we use a thread barrier so that all GPUs start updating a given row after having loaded its original, non-updated version in GPU memory. GPU memory requirements for Algorithm 3 are two block-columns (i.e. $2Nt^2$ numbers). As discussed above, rows are assigned to GPUs in a 1D block-cyclic way. Such allocations are recorded in the `blockAllocs` variable.

An adaptation of Algorithm 3 is possible when in-place operation is not needed: it is sufficient to remove the synchronization barrier, and change line 18 to write the output to a different matrix.

**Cholesky decomposition.** We want to decompose positive definite matrix $A$ into lower triangular matrix $L$ such that $L^\top L = A$. But $A$ does not fit entirely in GPU memory, and potentially more than one GPU is available. As before it is convenient to subdivide $A$ into smaller tiles such that the tiles fit

**Algorithm 4** Out-of-core, in-place Cholesky decomposition of symmetric positive definite matrix $A$. The lower triangle of $A$ will be overwritten by $L$ such that $L^\top L = A$. The function `OocPotrf` should be called for each available GPU with different values of the `procId` variable to parallelize the decomposition across GPUs. The inputs are the same as for Algorithm 3 but for work-table $T \in \mathbb{Z}^{N \times N}$ whose values are atomically updated by the different GPU processes to ensure synchronization.

```
 1: function OOCPOTRF(A, blockAllocs, procId, T, N)      35: function LOAD(A, T, i, j, exp)
 2:     for i = 1, ..., N do                             36:     while T_{i,j} < exp do
 3:         if i ∈ blockAllocs[procId] then              37:         wait
 4:             B ← Load(A, T, i, j, i)                   38:     end while
 5:             B ← POTRF(B)                              39:     return ToGPU(A_{i,j})
 6:             A_{i,i} ← Write(B, T, i, i)               40: end function
 7:         end if
 8:         for j ∈ blockAllocs[procId] do               41: function WRITE(G, T, i, j)
 9:             if j ≤ i then                             42:     T_{i,j} ← T_{i,j} + 1
10:                 continue                              43:     return FromGPU(G)
11:             end if                                    44: end function
12:             B ← Load(A, T, i, i, i + 1)
13:             C ← Load(A, T, j, i, i)
14:             C ← C(B^{-1})^⊤              ▷ via TRSM
15:             A_{j,i} ← Write(C, T, j, i)
16:         end for
17:         for j ∈ blockAllocs[procId] do
18:             if j ≤ i + 1 then
19:                 continue
20:             end if
21:             C ← Load(A, T, j, i, i + 1)
22:             for y = i, ... j do
23:                 E ← Load(A, T, j, y, i)
24:                 if y = j then
25:                     E ← E − CC^⊤         ▷ via SYRK
26:                 else
27:                     D ← Load(A, T, y, i, i + 1)
28:                     E ← E − DC^⊤         ▷ via GEMM
29:                 end if
30:                 A_{j,y} ← Write(E, T, j, y)
31:             end for
32:         end for
33:     end for
34: end function
```

in GPU memory.

$$\begin{pmatrix} A_{1,1} & & & \\ A_{2,1} & A_{2,2} & & \\ \vdots & & \ddots & \\ A_{n,1} & \ldots & & A_{n,n} \end{pmatrix} = \begin{pmatrix} L_{1,1} & & & \\ L_{2,1} & L_{2,2} & & \\ \vdots & & \ddots & \\ L_{n,1} & \ldots & & L_{n,n} \end{pmatrix} \begin{pmatrix} L_{1,1}^T & L_{2,1}^T & \ldots & L_{n,1}^T \\ & L_{2,2}^T & \ldots & L_{n,2}^T \\ & & \ddots & \vdots \\ & & & L_{n,n}^T \end{pmatrix}$$

Then the in-place decomposition may proceed column-wise across matrix $A$, where each column update requires three steps. The first step is to use the in-core `POTRF` function from cuSOLVER [34] on a single tile. Then, a triangular solution step is used to update the remaining rows of the first column (taking the first column as an example $A_{j,1} = L_{j,1}L_{1,1}^\top, 1 < j < N$, so clearly $L_{j,1} = A_{j,1}(L_{1,1}^{-1})^\top$). This can be done by using the `TRSM` operation from any GPU BLAS implementation. Finally, the *trailing* submatrix must be updated with those terms which can be computed from the current column, so that after this last step such column is not needed anymore. This step consists of running $A_{ij} = A_{ij} - L_{i,1}L_{j,1}^\top$ where if $c$ is the current column $i > c, \quad c < j \leq i$ (refer to Figure 5 for a more intuitive picture).

Running this algorithm in parallel requires dealing with several data dependencies in-between tiles, and in general it will not be possible to achieve perfect parallelism due to the inherently serial step of

performing the Cholesky decomposition of the first tile in a column. We avoid coarse synchronization mechanisms such as the thread barrier which was used for the LAUUM OOC implementation, since we found they could introduce very high waiting times (barriers would be needed after each of the three steps of the algorithm to ensure proper synchronization). Our solution, which somewhat follows [28], uses an integer table $T$ with one entry per tile, which is shared between all GPU threads. The entries of $T$ represent the current state of each tile: basically how many times the tile has been updated. Since we use a static row-cyclic work allocation like for the triangular matrix multiplication, each thread knows the expected state of a tile for each step (e.g. to perform the first step on tile $A_{c,c}$ the tile must have been updated exactly $c$ times). So it can wait until such state has been reached, then read the tile into GPU memory, perform the update, write back the tile to main RAM, and increment the corresponding entry in $T$. Such a scheme is implemented in Algorithm 4 with the help of the `Load` and `Write` sub-routines. Further optimizations are possible by being careful about which tiles are swapped in and out of GPU memory and at what times, overlapping computation with memory transfers when possible. Such optimizations generally require to increase the total memory allocated on the GPU, thus decreasing the maximum possible tile-size.