[Reviews · NeurIPS 2020]

Review 1

Summary and Contributions: In this paper, the authors introduce specialized and well engineered software for kernel ridge regression. The idea is to apply iterative numerical solvers to the Nystrom approximation, but with special care given to considerations like out of core computation, efficient memory usage and allocation, and so on.

Strengths: This is a solid machine learning systems paper, with appropriate care given to the various engineering challenges. I think engineering efforts like this are fantastic, and the empirical results for kernel ridge regression are impressive.

Weaknesses: In my opinion, comparing Nystrom for kernel ridge regression to variational GPs is apples to oranges in a lot of ways that are frankly unfair to variational GPs. In my view, a much more appropriate comparison would be a KeOps based implementation of SGPR or FITC with fixed inducing points. Variational GPs introduce a very large number of parameters in the form of the variational distribution and inducing point locations that require optimization and significantly increase the total amount of time spent in optimization. Methods that train GPs through the marginal likelihood with fixed inducing locations (e.g., as in Nystrom) may have as few as 3 parameters to fit. By contrast, SVGP learns (1) a variational distribution q(u) including a variational covariance matrix, and (2) the inducing point locations. These things result in a much more general version of the model, and are done to target test log likelihood, but increase the number of parameters from a handful to thousands or even millions in the full covariance S case. The effect of even subtle variations in these parameterizations have on wall clock efficiency both in terms of convergence rate and iteration speed is enormous and isn't considered. In fact, the effect of this can be seen in your own results -- you use GPyTorch 1.0.1, which is old enough to not have used a properly whitened parameterization of the variational parameters, while current versions of both GPFlow and GPyTorch do. These parameterization considerations can easily lead to several factors of speed up. The point of the above is not to nitpick specific package versions or various settings that could be used, but rather to argue that I don't believe SVGP is a fair baseline to compare a Nystrom based KRR solver to. More appropriate would be a standard KeOps based implementation of SGPR or FITC, with and without learning the inducing point locations, all of which are already supported by the relevant software.

Correctness: Again, my main point of concern is with the empirical methodology, specifically in the choice of SVGP as a baseline for Nystrom based KRR.

Clarity: The paper is very clearly written, and I have no comments here.

Relation to Prior Work: The authors do a good job of describing some relevant work, although my main source of criticism involves the focus on a comparison of Nystrom and SVGP, when I think there are more appropriate points of comparison.

Reproducibility: Yes

Additional Feedback: I want to emphasize here that I really do appreciate the sheer engineering effort that must have gone in to developing the software underlying this paper and am willing to consider accepting the paper regardless of my points above. From a scientific perspective, I just don't currently believe that Nystrom based KRR / predictive-mean-only GP and SVGP are at all comparable methods, although I am willing to be convinced otherwise. SVGP targets a much more general form of model, and pays a heavy parameterization price for it. I appreciate that you have at least used diagonally constrained covariance matrices for SVGP, since in the full variational covariance S case more than 99% of the parameters would be devoted to a predictive term that you aren't using. Nevertheless, I still think that approximate methods like FITC / SGPR trained with KeOps targeting the marginal likelihood would be much more appropriate points of comparison. I believe it is likely that your approach will significantly outperform a simple KeOps based solver for high dimensional datasets. --------- In the discussion period, I was in the minority for voting below the threshold. I am certainly not willing to argue strongly for the paper's rejection, as I mention above I quite like the methodology and appreciate the sheer engineering effort involved. I do strongly maintain my stance that using SVGP as a primary baseline is not useful. I appreciate the authors' point in the rebuttal that they point out the additional advantages of SVGP, but I feel that nevertheless using it as a primary point of comparison is still apples to oranges for the reasons I mention. Optimization considerations like whitening make enormous impacts on SVGP as a result of the sheer number of parameters involved -- particularly with a large number of inducing points -- and I would urge the authors to consider these factors more fairly than they currently do.


Review 2

Summary and Contributions: The computations involved in kernel methods must be done efficiently to scale kernel methods to large datasets. This paper proposes and implements a more computationally efficient version of FALKON, a kernel method by Rudi et al., 2017 that is practically statistically optimal based on the Nystrom method. This paper's main contribution is their highly-optimized implementation of FALKON that runs almost 20x faster than the version implemented by Rudi et al., 2017. The main algorithmic ideas of FALKON — subsampling points to make the Nystrom approximation, using iterative methods to reduce computational burden, using a carefully designed preconditioner that accelerates convergence, etc. — remain the same, but the authors carefully piece these ideas together to minimize memory usage and maximize computational throughput. This is done by using equivalent expressions that minimize the required memory, implementing a distributed, out-of-core, in-place Cholesky decomposition, and pipelining computation to overlap computation and data-transfer. The paper also derives a version of FALKON for loss functions beyond squared loss in the appendix. The paper compares their FALKON implementation with squared and logistic loss to other packages designed for large-scale kernel methods, including EigenPro, GPyTorch, and GPFlow. The results show large speedups despite modest hardware and demonstrating that proper engineering can make FALKON viable for even billions of data points. ************************** Post rebuttal: I thank the authors for their rebuttal. It is clear that the implementation is highly optimized. The rebuttal and promised edits addressed my concerns and I will raise my score to a 7. **************************

Strengths: The biggest strength of the paper is in the careful considerations of how to better implement FALKON by combining in-place Cholesky decompositions, GPU acceleration, state-of-the-art libraries like KeOps, preconditioning, and others. This allows the paper to use the Nystrom kernel approximation on over a billion data points with only 2 GPUs and a reasonable amount of RAM. The authors also took care to evaluate performance on a series of representative tasks, including sparse data and high-dimensional data. As large-scale machine learning pushes the boundaries of our computational limits, clever optimizations such as the ones presented in the paper enable practitioners to do more with the same computational resources, which is relevant for the community.

Weaknesses: Although the optimizations of the paper are sound and impressive, the empirical evaluations could be improved. Given that the main contribution of the paper is in building a highly optimized library, the presented benchmarks for efficiency should be as strong as possible. I appreciate the comparison to other packages designed for large scale kernel methods. However, Table 2 should clarify that the results for GPyTorch and GPFlow correspond to stochastic variational Gaussian processes (SVGPs). Currently, this is only mentioned in the main text away from the table. The same can be said about Figure 1. Also, does "time" column in Table 2 correspond to total training time, total prediction time, or both? Similarly, in Figure 1, what does the x-axis measure? Given that the training procedure for SVGP, stochastic optimization in minibatches, is drastically different from that of FALKON, solving a linear system in the case of squared loss, comparing training times without any qualifications would be potentially misleading. I would recommend comparing the training time for FALKON against just EigenPro, the most similar method, but comparing prediction times for FALKON against all the kernel methods. The authors should still report the training times for SVGPs but make clear how the training procedures differ from FALKON. If it is possible with the proposed implementation, the authors may want to include results on how the performance scales beyond 2 GPUs. Currently the appendix only shows the scaling with 1 and 2 GPUs.

Correctness: The theoretical claims are correct but can be hard to understand at places. However, the empirical methodology could be improved as I mention in the "Weaknesses" section.

Clarity: While the paper provides mostly clear exposition, the writing in the methodology sections can be improved. For example, much of section 3 mentions computing kernel matrices "in blocks", e.g. line 100, line 155, line 157, etc. However, it is not clear what this means. Are those matrices divided into blocks and computed one at a time, possibly in parallel? Similarly, the explanation around Figure 4 is also confusing. The allocated matrix is size m x m with the upper half storing an upper triangular T. However step (d) claims to store TT^\top, a m x m matrix, in the lower half of the allocated memory which is not enough for the m x m matrix.

Relation to Prior Work: Yes the paper makes clear that they are optimizing the computations required for a prior method.

Reproducibility: Yes

Additional Feedback: line 156 "in-place Cholesky decomposition is performed on the GPU" Which matrix is being decomposed? line 158: "Cholesky is performed on the lower triangle" Which matrix is being decomposed? line 171: "q x m" Based on figure 3, should this be "q x d"?


Review 3

Summary and Contributions: I have read the rebuttal and other reviewer's comments. I appreciate the authors' reply. I would like to keep my score on this work. ========================== This paper integrates several practical techniques to further scale the training of kernel machines. Its algorithm is based on a preconditioned conjugate gradient solver proposed in [43]. The major techniques proposed in this work are to relax memory constraint (of GPU) by performing low precision out-of-core computation and utilize more computation power by distributing computation among several GPUs. The final implementation shows satisfying speedup over several state-of-the-art kernel methods with highly competitive performance.

Strengths: I tend to accept this work as it presents solid algorithmic and system improvements for the state-of-the-art kernel methods.

Weaknesses: As space used by the preconditioner is quadratic in the number of Nystrom centers, it seems challenging to further increase the size of the kernel model, e.g., beyond 10^6 centers. What would be the possible solutions? Adopting float32 over float64 gives substantial savings on memory without much loss in performance. Do the authors think that the precision can be further dropped to float16 or even one byte (similar to what deep net uses nowadays)? If not, what are the major obstacles? I noticed that in the machine for experiments is equipped with two 12-core CPUs. So would CPU be the bottleneck when training with a consumer desktop (which typically has one cpu socket)? It would be nice if the authors can also present CPU and GPU utilization during training.

Correctness: Both are correct.

Clarity: It is very well organized and easy to follow.

Relation to Prior Work: Yes

Reproducibility: Yes

Additional Feedback:


Review 4

Summary and Contributions: Kernel methods provide an elegant and principled approach to nonparametric learning, but so far could hardly be used in large scale problems, since naïve implementations scale poorly with data size. Recent advances have shown the benefits of a number of algorithmic ideas, for example combining optimization, numerical linear algebra and random projections. Here, we push these efforts further to develop and test a solver that takes full advantage of GPU hardware. Towards this end, we designed a preconditioned gradient solver for kernel methods exploiting both GPU acceleration and parallelization with multiple GPUs, implementing out-of-core variants of common linear algebra operations to guarantee optimal hardware utilization. Further, we optimize the numerical precision of different operations and maximize efficiency of matrix-vector multiplications. As a result we can experimentally show dramatic speedups on datasets with billions of points, while still guaranteeing state of the art performance. Additionally, we make our software available as an easy to use library.

Strengths: The strength lies in the fact that a number of carefully design computational approaches specifically adapted to the GPU were developed for solving SVM using the Nystrom method. The resulting Falkon algorithm developed in the paper is efficient and the evaluation is sound. It is of extreme relevance to the NeurIPS community as this line of works stands in the trend of tweaking neural networks for performance gains, adding much needed diversity to the pool of thoughts.

Weaknesses: No much obvious weakness. Only nitpicking.

Correctness: The claims and methods are provably correct. The empirical methodology is well thought out.

Clarity: The paper is well written.

Relation to Prior Work: Kernel methods used for large scale problems are the most related line of work. In this paper, the preconditined conjugate gradient solvers that take advantage of GPUs were compared with EigenPro, GPytorch, and GPflow. The paper could use more description of the close relative of EigenPro.

Reproducibility: Yes

Additional Feedback: see sections above

[Author Response · NeurIPS 2020]

Thanks to all reviewers for their careful reading and thoughtful comments.

**Reviewer #1**. The rationale in the choice of the comparisons for the experiments was to be as extensive as possible:
covering the methods which solve Nyström KRR first, then related methods such as those which solve kernel regression
with different smooth losses or other sketching techniques, and finally we wished to cover partially related kernel based
methods i.e. approaches computing similar quantities as the predictive mean of variational GPs (VGPs). Indeed, we
agree with Rev1 on the difference between Nyström KRR and VGPs, in terms of methods and objectives, as we already
clarified in Section 4 (there we also pointed out the greater generality of SVGP). To be more explicit we will further
clarify the scope of our experiments and the difference between Nyström KRR and VGPs in Section 4. From this
viewpoint we agree that FITC / SGPR with KeOps would better fit the third category than SVGP and we will include the
suggested methods in our experiments. Note that we had considered the option of comparing with FITC / SGPR while
designing the experiments. However, in the end we opted for SVGP since it can be trained in minibatches, allowing a
higher number of inducing points, contrary to the implementations of SGPR in GPyTorch and GPflow that require all
data and gradients to be stored on the GPU, thus strongly limiting the number of inducing points. For example on our
hardware, GPflow SGPR runs out of GPU memory with only 20 inducing points on the HIGGS dataset. This shows
that current implementations of SGPR cannot scale on big dataset. However, for the sake of comparison, we decided to
reduce the training set to $n = 300000$ points for each dataset. This allows SGPR to fit 1000 inducing points in memory.
We then ran one experiment with fixed SGPR points and the same number (1000) of Falkon centers, and another where
we trained the SGPR points, and ran Falkon by tuning the number of centers such that the computational time matched
that of training SGPR to convergence. In this second case we can compare the accuracy of the two methods. In Table 1
we provide some preliminary results on a subset of our datasets. Notice that in the first experiment Falkon is much
faster than SGPR, and the two methods have comparable accuracy; in the second experiment, the accuracy of Falkon is
higher or equal to that of SGPR. The final version of the paper will compare the two methods on all considered datasets.

**Reviewer #2**. As suggested by the reviewer we will clarify in Table 2 and in Figure 1 that the results for GPyTorch
and GPflow correspond to the SVGP algorithm. In particular, as suggested by Rev2 we will report the training times
of SVGP clarifying how the procedure differs from the one of Falkon. Indeed by "time" we mean *total training time*
since in general, this is the limiting factor. The crucial difference in the optimization of the two algorithms is due to
the convexity of the FALKON objective which allows us to use the more efficient preconditioned conjugate gradients
optimization, based on out-of-core matrix vector operations, instead of ADAM. We will recall this difference in the
experimental section, before Table 2. As suggested by Rev2 we will include a table on how performance scales with
more GPUs on the considered datasets. According with our current hardware availability, we will consider how the
performance scales with 1 to 4 GPUs (see Fig. 1 for the TAXI dataset). To conclude, with "blocks" we mean that
matrices are divided into blocks which fit in GPU memory. Each GPU will then be assigned to work on different subsets
of blocks which it will process one at a time. In Figure 4 we neglected to mention that $TT^\top$ is a symmetric matrix, so
we store only its lower triangular part.

**Reviewer #3**. We thank the reviewer for the thoughtful comments and interesting questions. We agree with Rev3, right
now Nyström methods based on preconditioning can scale up to $10^6$ centers since they are limited by the available
memory. This can be addressed in different ways: 1) achieving the same accuracy with fewer centers is possible by
picking better centers e.g. using leverage scores or DPP sampling instead of random sampling. 2) recomputing the
preconditioner on the fly, in blocks, at each iteration; this would remove the memory bottleneck, but would make the
algorithm $O(\log n)$ times slower 3) further sketching the preconditioner itself with randomized linear algebra. We will
add a remark with these considerations at the end of Section 5. Reducing the precision is also an interesting direction
to increase the number of Nyström points and to leverage the GPU and TPU architectures. The crucial aspect in this
direction is to guarantee numerical stability of the computation of the kernel and of Cholesky decomposition. Interesting
starting points in this direction are stochastic rounding techniques (see e.g. "Stochastic Rounding and its Probabilistic
Backward Error Analysis" of Connolly et al. 2020). To conclude, CPU power is not crucial to run our implementation
of FALKON since all the computations are done by GPU, while CPU essentially controls only the logic of the program.

|  | MSD-300k | | SUSY-300k | | HIGGS-300k | |
|---|---|---|---|---|---|---|
|  | time | MSE | time | AUC | time | AUC |
| GPflow SGPR (fixed 1k points) | 7s | 83.06 | 8s | **87.43** | 6s | 74.48 |
| Falkon (the same 1k points) | 1.5s | **83.01** | 1s | 87.39 | 0.4s | **75.58** |
| GPflow SGPR (trained points) | 40s | 77.24 | 24s | 87.49 | 11s | 77.35 |
| Falkon (match time) | 42s | **76.15** | 19s | **87.51** | 9s | **77.66** |

Figure 1: multi-GPU scaling of the TAXI dataset (experiment run on the workstation Nvidia DGX-Station).

Table 1: Falkon is faster and more accurate than SGPR in the first and third setting; in the second, with very few inducing points, Falkon is much faster but has slightly worse accuracy.

[Meta-Review · NeurIPS 2020]

There is a consensus among the knowledgeable reviewers that this work makes a significant contribution to the kernel community. It integrates several practical techniques and engineering efforts to further improve the scalability of the kernel machines. The techniques proposed in this work will permit the use of several GPUs in training kernel-based models with huge amount of data, which I also see as a significant contribution. Regardless of the overall score, I think this paper deserves an oral because it shows how to take full advantage of GPU hardware when solving learning problems with kernels methods. Scalability is one of the long-standing problems in kernel machines but has been largely neglected and under-appreciated in the past few years. Unlike previous work, this paper presents a solution based on non-trivial engineering efforts such as out-of-core implementation that allow us to exploit both GPU acceleration and parallelization with multiple GPUs. This development could open up new applications for kernel methods in areas that have previously been dominated by deep learning models. The paper is therefore accepted as an oral presentation. Last but not least, R1 and R2 raised an important concern regarding the empirical comparison to SVGP. I hope that the authors will take reviewers' comments into account when revising the manuscript for the camera-ready version by improving the justification of their empirical comparison.